# An Antisense Oligonucleotide against a Splicing Enhancer Sequence within Exon 1 of the *MSTN* Gene Inhibits Pre-mRNA Maturation to Act as a Novel Myostatin Inhibitor

**DOI:** 10.3390/ijms23095016

**Published:** 2022-04-30

**Authors:** Kazuhiro Maeta, Manal Farea, Hisahide Nishio, Masafumi Matsuo

**Affiliations:** 1KNC Department of Nucleic Acid Drug Discovery, Faculty of Rehabilitation, Kobe Gakuin University, Kobe 651-2180, Hyogo, Japan; maeta@reha.kobegakuin.ac.jp (K.M.); manalr44@reha.kobegakuin.ac.jp (M.F.); 2Research Center for Locomotion Biology, Kobe Gakuin University, Kobe 651-2180, Hyogo, Japan; nishio@reha.kobegakuin.ac.jp; 3Department of Occupational Therapy, Faculty of Rehabilitation, Kobe Gakuin University, Kobe 651-2180, Hyogo, Japan

**Keywords:** antisense oligonucleotide, splicing, splicing enhancer sequence, myostatin

## Abstract

Antisense oligonucleotides (ASOs) are agents that modulate gene function. ASO-mediated out-of-frame exon skipping has been employed to suppress gene function. Myostatin, encoded by the *MSTN* gene, is a potent negative regulator of skeletal muscle growth. ASOs that induce skipping of out-of-frame exon 2 of the *MSTN* gene have been studied for their use in increasing muscle mass. However, no ASOs are currently available for clinical use. We hypothesized that ASOs against the splicing enhancer sequence within exon 1 of the *MSTN* gene would inhibit maturation of pre-mRNA, thereby suppressing gene function. To explore this hypothesis, ASOs against sequences of exon 1 of the *MSTN* gene were screened for their ability to reduce mature *MSTN* mRNA levels. One screened ASO, named KMM001, decreased *MSTN* mRNA levels in a dose-dependent manner and reciprocally increased *MSTN* pre-mRNA levels. Accordingly, KMM001 decreased myostatin protein levels. KMM001 inhibited SMAD-mediated myostatin signaling in rhabdomyosarcoma cells. Remarkably, it did not decrease *GDF11* mRNA levels, indicating myostatin-specific inhibition. As expected, KMM001 enhanced the proliferation of human myoblasts. We conclude that KMM001 is a novel myostatin inhibitor that inhibits pre-mRNA maturation. KMM001 has great promise for clinical applications and should be examined for its ability to treat various muscle-wasting conditions.

## 1. Introduction

We first proposed the use of antisense oligonucleotide (ASO)-mediated exon-skipping therapy for Duchenne muscular dystrophy (DMD) in 1995 [1,2]. ASO-mediated exon skipping is intended to convert out-of-frame *DMD* mRNA into in-frame *DMD* mRNA, enabling the production of dystrophin, which is deficient in DMD [2]. Since our proposal, an ASO designed to target an exonic splicing enhancer sequence has been shown to induce exon skipping [3]. ASO-mediated exon skipping has since developed into a verified strategy for a subset of patients with DMD [4]. As exon-skipping therapy enables mRNA editing, it has been applied to treat both monogenic and nonmonogenic diseases [5]. Notably, ASO-mediated splicing can be applied to manipulate gene functions not only for restoration but also for destruction and modulation [5]. For example, splice-modulating ASOs that target intronic splicing silencer sequences are currently being used to treat patients with spinal muscular atrophy [6]. In these cases, internal exons flanked by both splice acceptor and donor sites are modulated by the ASOs.

Splicing is a step in which introns are excised from pre-mRNAs that are transcribed from genes to produce mature mRNAs. To ensure accurate mRNA production, exons are defined by splicing consensus sequences located at the junctions of exons and introns in pre-mRNAs. GT and AG dinucleotides are highly conserved consensus sequences at the splice donor and acceptor sites, respectively. Additionally, the definition of exons is supported by splicing cis elements, such as splicing enhancer and silencer sequences located within exons or introns. If a nucleotide change is present in these cis-regulatory sequences, a splicing error may occur [7]. In the case of DMD treatment, ASOs have been designed against exonic splicing enhancers to induce exon skipping [2]. Exonic splicing enhancers serve as binding sites for specific serine/arginine-rich (SR) proteins, a family of structurally related and highly conserved splicing factors named SRSF1 to SRSF12 [8]. Splicing generally proceeds without errors because of a sophisticated quality control system.

Myostatin, also known as growth differentiation factor 8 (GDF8), is a member of the transforming growth factor β (TGF-β) superfamily and a potent negative regulator of skeletal muscle growth [9,10]. It is encoded by the *MSTN* gene, which comprises three exons and spreads over 7 kb on chromosome 2 [11]. Myostatin, a secretory protein, is synthesized in skeletal muscle as a precursor that undergoes maturation steps [12,13]. The inactive precursor (pro-myostatin) comprises 375 amino acid residues, with an N-terminal signal peptide, a prodomain, and a C-terminal active growth factor domain [10]. The active domain binds and activates receptors on the surfaces of muscle cells. Subsequently, a SMAD-dependent complex is formed and translocated into the nucleus, where the complex activates genes that drive muscle wasting [14]. Although multiexon genes are usually subjected to alternative splicing, alternative splicing was previously believed not to occur in the human *MSTN* gene. Recently, we identified alternative splicing of the *MSTN* transcript for the first time (deposited in GenBank under accession number MZ436933). This finding indicates that the *MSTN* gene is not excluded from common splicing regulatory mechanisms.

Knocking out the *MSTN* gene in mice results in a two- to three-fold increase in muscle mass [9]. Mutations in the *MSTN* gene have been reported to produce the “double-muscled” phenotype in animals, such as mice, cattle, sheep, and dogs [15,16,17,18]. Consistent with the high conservation of the *MSTN* gene among animals [10], a mutation in the human *MSTN* gene has been found to produce markedly increased musculature in a child [19]. Therefore, myostatin inhibition has become an attractive prospective strategy for increasing muscle mass in muscle-wasting conditions, such as muscular dystrophy, sarcopenia, and cancer-associated cachexia [20,21]. ASOs that can induce the skipping of out-of-frame exon 2 of the *MSTN* gene in order to suppress *MSTN* gene function have been studied [22,23]. However, no ASOs have been officially approved for clinical use [24]. Thus, the development of clinically effective myostatin inhibitors is still awaited.

The first exon at the 5′ end of pre-mRNA is defined by the splice donor site only. Although it is common to identify exon skipping due to mutations in internal-exon splicing regulatory elements [5], the splicing outcomes induced by mutations in exon 1 splicing regulatory elements are not well understood. Recently, mutations in the splice donor site of exon 1 have been reported to result in retention of intron 1 in mRNA [25]. Given these findings, we hypothesized that ASOs against exonic splicing enhancers in exon 1 of the *MSTN* gene would inhibit splicing and decrease mature mRNA levels.

Here, we tested this hypothesis by screening ASOs that inhibit the production of mature *MSTN* mRNA. Chimeric ASOs consisting of 2′-*O*-methyl RNA (2′-OMeRNA) and 2′-*O*,4′-*C*-ethylene-bridged nucleic acid (ENA^®^) were designed against exonic splicing enhancer sequences within exon 1 of the *MSTN* gene and were examined for their ability to reduce *MSTN* mRNA levels in CRL-2061 rhabdomyosarcoma cells that highly expressed *MSTN* mRNA. One ASO, named KMM001, decreased the mRNA levels in a dose-dependent manner and reciprocally increased the pre-mRNA levels. As expected, KMM001 decreased myostatin expression and inhibited myostatin signaling in CRL-2061 cells. Remarkably, it did not decrease *GDF11* mRNA levels, indicating myostatin-specific inhibition. In addition, as expected, KMM001 enhanced the proliferation of human myoblasts. We conclude that KMM001 is a novel myostatin inhibitor that inhibits pre-mRNA maturation. KMM001 has great promise for clinical applications and should be examined for its ability to treat various muscle-wasting conditions.

## 2. Results

### 2.1. Identification of a Chimeric ASO That Decreases Mature MSTN mRNA Levels

We hypothesized that an ASO against the splicing enhancer sequence within exon 1 of the *MSTN* gene would inhibit splicing of intron 1 of *MSTN* pre-mRNA. To design the ASO, the nucleotide sequence of *MSTN* exon 1 was examined for a splicing enhancer sequence with ESEfinder3.0. As expected, the binding sites for SRSF1, SRSF1 (IgM-BRACA1), SRSF2, SRSF5, and SRSF6 were identified in exon 1 and were clustered in some locations (Figure 1).

As the first step of screening, three clusters were selected as candidates for the ASO target site. Then, 18-mer chimeric ASOs constituting 2′-OMeRNA and ENA nucleotides were synthesized against three locations (AO1, AO2, and AO3) (Figure 1B) and were transfected into CRL-2061 cells, a cell line established from a human rhabdomyosarcoma. After 24 h of transfection, *MSTN* mRNA was semiquantified by RT–PCR amplification of a fragment extending from exons 1 to 3, and glyceraldehyde 3-phosphate dehydrogenase (*GAPDH*) mRNA was also amplified as a control. Then, the *MSTN/GAPDH* ratio was calculated from the density to evaluate the activity. All three ASOs decreased the ratio, but AO2 produced the lowest ratio (Figure 2A). To identify more effective ASOs, five additional ASOs shifted 3 bp upstream or downstream of AO2 were synthesized (Figure 2B) and examined for their effect on *MSTN* mRNA production. Two AOs that shifted 3 and 6 bp upstream from AO2 (AO2+3 and AO2+6, respectively) decreased the *MSTN/GAPDH* ratio more strongly than AO2, with the effect of AO2+3 being the strongest (Figure 2C). However, two ASOs designed 3 and 6 bp downstream from AO2 did not decrease the *MSTN/GAPDH* ratio more strongly than AO2. Since AO2+3 seemed to be a suitable sequence, another four ASOs that were shifted 1 bp downstream or upstream from AO2 or AO2+3 were synthesized in the third screening (Figure 2B). Remarkably, AO2+1 decreased the *MSTN/GAPDH* ratio most strongly, causing it to reach nearly 0.1 (Figure 2D). To ensure that no other ASOs decreased the ratio more than AO2+1, ASOs were designed against 12 locations scattered over exon 1, and their splicing products were analyzed (Figure 2E). However, no ASO decreased the ratio more strongly than AO2+1. Thus, AO2+1 was concluded to be the ASO most suitable for decreasing *MSTN* mRNA levels.

### 2.2. Amelioration of AO2+1-Induced Splicing Inhibition by SRSF5

To characterize the splicing regulatory proteins involved in AO2+1-mediated inhibition, the sequences around the complementary region of AO2+1 were analyzed with SpliceAid2. The results revealed an SRSF5 binding site at the 5′ end of the sequence complementary to AO2+1 (Figure 3). Since AO2+1 was suggested to inhibit the binding of SRSF5, SRSF5 was overexpressed in AO2+1-treated cells. In the absence of AO2+1, SRSF5 expression did not markedly affect the *MSTN/GAPDH* ratio. However, surprisingly, in AO2+1-treated cells, SRSF5 expression increased the *MSTN/GAPGH* ratio in a dose-dependent manner, causing the ratio to reach 0.8 at a dose of 1 μg of plasmid (Figure 3). These results indicate that AO2+1 inhibits splicing by hindering the function of SRSF5 as the splicing enhancer sequence.

### 2.3. Determination of the Best Structure of AO2+1

AO2+1 is a chimeric ASO constituting 2′-OMe RNA and ENA. Therefore, the number and location of ENA within the ASO was suspected to affect the activity of AO2+1. This possibility was examined by replacing ENA with 2′–OMe RNA (to avoid long repeats of ENA) and synthesizing three ASOs (AO2+1a, AO2+1b, and AO2+1c). These ASOs were examined for their ability to decrease the *MSTN/GAPDH* ratio in CRL-2061 cells. However, none of the three ASOs decreased the *MSTN/GAPDH* ratio more than AO2+1 (Figure 4). Therefore, AO2+1 was identified as the most suitable ASO structure. This ASO was named KMM001 and further studied for its drug development potential.

### 2.4. KMM001 Decreased Mature mRNA Levels in a Dose-Dependent Manner and Reciprocally Increased Pre-mRNA Levels

To confirm the splicing inhibition mediated by KMM001, KMM001 was added to the culture medium of CRL-2061 cells at concentrations ranging from 0 to 200 nM. The expression levels of mature *MSTN* mRNA and pre-mRNA were analyzed by RT–PCR amplification of a fragment extending from exon 1 to 3 and a fragment extending from intron 2 to exon 3, respectively (Figure 5). The *MSTN/GAPDH* and pre-*MSTN/GAPDH* ratios were calculated from the amplified band densities. When the KMM001 concentration was increased from 0 to 100 nM, the *MSTN*/*GAPDH* ratio decreased dose dependently (Figure 5). However, the pre-*MSTN/GAPDH* ratio increased dose dependently when the KMM001 concentration was increased from 0 to 100 nM (Figure 5). These results indicated the occurrence of reciprocal changes in *MSTN* mature mRNA and pre-mRNA. To further confirm intron retention, the full length of intron 1 was amplified under 200 nM KMM001. As expected, the 2.4 kb fragment extending from exon 1 to exon 2 was amplified from the transfected cells, whereas no product was obtained from the nontransfected cells (Appendix A). To exclude the possibility of contamination by genomic DNA, the genomic region of exon 45 of the *DMD* gene was PCR amplified. However, the amplified product was not produced in any of the samples (Appendix A). These findings indicate that KMM001 inhibits the splicing of intron 1 of *MSTN* pre-mRNA.

### 2.5. KMM001 Did Not Decrease GDF11 mRNA Levels

Given the high homology between the *MSTN* gene and the *GDF11* gene, a member of the TGF-β superfamily, the generation of myostatin-specific inhibitors was especially difficult. The lack of specificity has the potential to lead to unwanted side effects [26]. It was feared that KMM001 would additionally decrease the amount of *GDF11* mRNA. However, in silico analysis of the *GDF11* gene sequence did not reveal any regions that were highly homologous to the complementary sequence of KMM001 (Appendix A). To further confirm this, KMM001 was added to the culture medium of CRL-2061 cells. Both *MSTN* and *GDF11* mRNA levels in the treated and nontreated cells were semiquantified by RT–PCR amplification (Figure 6). As expected, KMM001 decreased the *MSTN/GAPDH* ratio dramatically to 0.03, while the ratio was 1 in nontreated cells (*p* < 0.001). In contrast, the addition of KMM001 statistically insignificantly decreased the *GDF11/GAPDH* ratio to 0.87 (*p* = 0.29). This finding indicates that KMM001 preferentially decreases *MSTN* mRNA levels.

### 2.6. KMM001 Inhibited Myostatin Protein Expression in CRL-2061 Cells

We reasoned that the decreases in *MSTN* mRNA levels induced by KMM001 should reduce myostatin protein production. Thus, myostatin protein expression in CRL-2061 cells was examined by Western blot assay using an antibody against the N-terminal domain of myostatin. In nontreated CRL-2061 cells, a 75 kDa band was observed (Figure 7A). Remarkably, 100 nM KMM001 treatment for 48 h decreased the density of the band to 10% of the level in the nontreated cells, confirming that KMM001 inhibits myostatin expression at the protein level.

### 2.7. KMM001 Inhibited Endogenous Myostatin Signaling in CRL-2061 Cells

To determine the biological effect of KMM001, myostatin signaling was analyzed by assessment of SMAD-dependent luciferase activity in cells transfected with a SMAD−responsive reporter gene. The reporter gene was transfected into the cells together with the galactosidase plasmid as a control. After transfection of the reporter gene, luciferase activity was clearly observed in CRL-2061 cells but not in HEK293 cells (Appendix A). This indicated that endogenous myostatin activated SMAD−dependent luciferase. Therefore, this system was determined to be suitable to measure the signaling activity of myostatin. Then, KMM001 was added to the culture medium at concentrations from 1 to 200 nM (Figure 7B). KMM001 at concentrations ranging from 0 to 100 nM decreased the relative luciferase activity until a plateau was reached. KMM001 was therefore selected as a myostatin signaling inhibitor.

### 2.8. KMM001 Enhanced the Proliferation of Myoblasts

KM001 also reduced *MSTN* mRNA and protein level of myostatin in human myoblast (Appendix A). Given its myostatin-inhibiting function, KMM001 was predicted to enhance the proliferation of human myoblasts. To examine this possibility, KMM001 was added to the culture medium of immortalized human myoblasts. The cells were quantified by Cell Counting Kit-8 (CCK-8) assay and microscopic cell counting for 3 days. In the CCK-8 assay, the absorbance increased in treated and nontreated cells (Figure 8A). Notably, the absorbance at the third day was significantly higher in KMM001-treated cells than in nontreated cells (146%, *p* < 0.001), indicating that KMM001 enhanced cell proliferation. Direct cell counting using a fluorescence microscope showed that the numbers of human myoblasts increased over time. Notably, the number of KMM001-treated cells was significantly higher than that of nontreated cells at the third day (138%, *p* < 0.01) (Figure 8B). These results indicate that KMM001 is an enhancer of myoblast cell proliferation.

## 3. Discussion

In this study, KMM001, an 18-mer chimeric ASO complementary to the splicing enhancer sequence within exon 1 of the *MSTN* gene, was shown to inhibit the splicing of *MSTN* pre-mRNA. This effect was accompanied by a reciprocal increase in the amount of pre-mRNA. This was the first study to use an ASO to inhibit mRNA production, although ASOs have previously been used to modulate splicing and produce mRNAs with different exon compositions. Gene function has been silenced mostly via two techniques: RNase H-dependent degradation of mRNA directed by short chimeric ASOs and gene expression interference with short interfering RNAs (siRNAs) [27]. Our results add another technique for the silencing gene function, namely, ASO-mediated mRNA level reduction.

KMM001 is a chimeric ASO comprising 2′-OMe RNA and ENA. ENA, a kind of locked nucleic acid, is characterized by its high affinity for complementary RNA and its high stability [28]. Currently, an ASO comprising 2′-OMe RNA and ENA (Renadirsen) is under clinical trial for the treatment of DMD. Renadirsen has been reported to have a stronger ability to induce exon skipping in the skeletal and cardiac muscles of mice than ASOs comprising non-ENA monomers [29]. Therefore, KMM001 is strongly expected to inhibit myostatin in skeletal muscle and produce hypertrophic changes in atrophic muscles.

ASOs have been employed to activate RNase H in order to destroy mRNA or to switch splicing for mRNA editing [5]. Here, we applied an ASO to inhibit the maturation of pre-mRNA, hypothesizing that ASO-mediated steric inhibition of a splicing enhancer within exon 1 would inhibit the splicing of pre-mRNA. The roles of splicing enhancers within exon 1 are not well understood because splicing regulatory mechanisms have been studied mainly in internal exons that are defined by splice donor and acceptor sites. In this study, an ASO targeting the splicing enhancer sequence within exon 1 was shown to inhibit the splicing of intron 1. This finding indicates that recognition of the splice donor site of *MSTN* exon 1 requires support from cis-splicing regulatory elements, as is needed by internal exons. Our results open a new avenue for inhibition of the production of mature mRNA.

SRSF5 was identified as an SR protein that competed with KMM001 for binding to the target sequence. Overexpression of SRSF5 restored splicing to normal levels in KMM001-treated cells. The 18 nt target sequence of KMM001 encodes six amino acids of myostatin (the 36th to 41st residues). In Japanese quail, a deletion of the 3 nt encoding the 42nd cysteine residue of myostatin prevented myostatin protein production, even though it was expected that myostatin lacking one amino acid would be produced [30]. It is highly conceivable that in that case, the absence of myostatin protein was caused by a lack of production of mature mRNA due to disruption of the exonic splicing enhancer sequence. However, this supposition needs further confirmation. Such confirmation would strengthen the evidence regarding the roles of the splicing enhancer in the first exon.

Several biotherapeutic modalities, including anti-myostatin antibodies, a myostatin propeptide, a soluble ActRIIB-Fc, and ASOs that block signaling activity, have been demonstrated to inhibit myostatin [22,31,32,33]. As myostatin belongs to the TGF-β superfamily, the functional crossover occurs at the protein, receptor, and signaling levels. Therefore, the development of myostatin inhibitors that do not exert unwanted effects by inhibiting the functions of other members of the TGF-β superfamily, especially the *GDF11* gene (which is highly homologous with the *MSTN* gene), is important. Among previous studies on myostatin inhibitors, one clearly showed that an inhibitor inhibited GDF11 [34]; however, most studies have focused only on myostatin inhibition without considering GDF11 [35]. ASO treatment has been shown to inhibit splicing of exon 2 of the *MSTN* gene, thereby inducing exon skipping to produce out-of-frame mRNA [22]. Although this is a way to abolish the ability of mRNA to encode proteins, it is highly possible that splicing of the *GDF11* gene is also modulated, since the sequence of exon 2 of the *MSTN* gene is 80% homologous to *GDF11*. Assuming that this possibility is unavoidable as long as exon 2 is a target of an ASO for splicing modulation, we targeted exon 1, which is not as highly homologous with GDF11 as exon 2. We successfully found that KMM001 specifically inhibited *MSTN* mRNA production but not *GDF11* mRNA production.

Notably, multiple roles of myostatin have been elucidated in the heart [36] and vasculature [37] and in the contexts of insulin sensitivity [38,39], vascular remodeling [37], cell senescence, and fibrosis [40]. Since KMM001 can be administered subcutaneously like Renadirsen [29], it is more widely applicable than expected.

There were several limitations of this study. First, the effects of KMM001 were not examined in vivo using a mouse model. Thus, the in vivo effects need further study. Second, studies on SRSF5 have been limited, but further studies on SRSF5 will elucidate the mechanisms of splicing regulation of the *MSTN* gene. Third, differentiation of myoblasts was not analyzed and will be analyzed in the future study using different culture medium from the current medium.

## 4. Materials and Methods

### 4.1. Prediction of Exonic Splicing Enhancer Sequences

Exonic splicing enhancer sequences were predicted by ESEfinder3.0, a web-based program that identifies putative exonic splicing enhancers responsive to the human SR proteins SRSF1 (SF2/ASF), SRSF1 (SF2/ASF, IgM-BRCA1), SRSF2 (SC35), SRSF5 (SRp40)), and SRSF6 (SRp50) (http://krainer01.cshl.edu/cgi-bin/tools/ESE3/esefinder.cgi?process=home, accessed on 27 August 2018). SpliceAid 2, another web-based program that identifies putative exonic splicing enhancers and silencers responsive to human SR proteins (http://193.206.120.249/splicing_tissue.html, accessed on 3 September 2018), was also used to predict exonic splicing enhancers and silencers.

### 4.2. Cells

The rhabdomyosarcoma cell line CRL-2061 and the cell line HEK293 were purchased from ATCC (Manassas, VA, USA). CRL-2061 cells were cultured in RPMI 1640 medium (Nacalai Tesque, Inc., Kyoto, Japan) supplemented with 10% fetal bovine serum (FBS) (Gibco by Life Technologies, Grand Island, New York, NY, USA) and 1% antibiotic-antimycotic solution (Nacalai Tesque, Inc.) at 37 °C in a 5% CO_2_ humidified incubator. HEK293 cells were cultured in DMEM (High Glucose) (Nacalai Tesque, Inc.) supplemented with 10% FBS and 1% antibiotic-antimycotic solution at 37 °C in a 5% CO_2_ humidified incubator.

Human myoblasts that were generated by immortalization of primary cultured human myogenic cells and used in a myoblast proliferation study were a kind gift from Dr. Hashimoto. The myoblasts were grown in DMEM (High Glucose) supplemented with 20% FBS, 2% Ultroser G (Pall Life Sciences, Washington, NY, USA) and 1% antibiotic-antimycotic solution (Nacalai Tesque, Inc.) at 37 °C in a humidified incubator with 5% CO_2_.

### 4.3. Transcript Analysis

Cells (2 × 10^5^) grown on 12-well culture dishes were transfected with ASO in 4 μL of Lipofectamine 3000 (Thermo Fisher Scientific, Inc., Carlsbad, CA, USA). After incubation for 24 h, the cells were rinsed twice with phosphate-buffered saline (Nacalai Tesque, Inc.) and then collected using the Lysis/Binding Buffer of a High Pure RNA Isolation Kit (Roche Diagnostics, Basel, Switzerland). RNA was extracted using a High Pure RNA Isolation Kit (Roche Diagnostics). cDNA was synthesized from 0.5 µg of total RNA from each sample using random primers. The *MSTN* transcript was PCR amplified using sets of primers for exon 1 and exon 3 (MSTN Ex1F1: 5′-agattcactggtgtggcaag-3′ and MSTN R2: 5′-tgcatgacatgtctttgtgc-3′, respectively), exon 1 and exon 2 (MSTN Ex1F1: 5′-agattcactggtgtggcaag-3′ and MSTN Ex2R3: 5′-gttgtaggagtctcgacgggtct-3′, respectively), and intron 2 and exon 3 (MSTN In2F: 5′-caggcaatctggtactcaaac-3′ and MSTN Ex3R: 5′-cgtgattctgttgagtgctcat-3′, respectively). The *GDF11* transcript was PCR amplified using a set of primers for exon 1 and exon 3 (GDF11 Ex1F4: 5′-ctgcagcagatcctggacct-3′ and GDF11 Ex3R4: 5′-catgaacatgtactcgcact-3′, respectively). The *DMD* gene was PCR amplified using a set of primers for intron 44 and intron 45 (gDNA primer 45F: 5′-tgccagtacaactgcatgtggtag-3′ and gDNA primer 45R: 5′-gcttataatctctcatgaaatattc-3′, respectively). The integrity and concentration of the cDNA were examined by amplifying the mRNA of the housekeeping gene *GAPDH*.

PCR amplification was performed in a total volume of 20 µL containing 2 µL of cDNA, 2 µL of 10× ExTaq buffer (Takara Bio, Inc., Shiga, Japan), 0.25 U of ExTaq polymerase (Takara Bio, Inc.), 500 nM of each primer, and 200 µM dNTPs (Takara Bio, Inc.). For *MSTN*, *GDF11,* and *DMD* amplification, 30 cycles of amplification were performed on a Mastercycler Gradient PCR machine (Eppendorf, Hamburg, Germany) using the following conditions: initial denaturation at 94 °C for 3 min, subsequent denaturation at 94 °C for 0.5 min, annealing at 60 °C for 0.5 min, and extension at 72 °C for 1.5 min. For *GAPDH* amplification, 18 cycles of amplification were performed. Amplified PCR products were electrophoresed on agarose gels and semiquantified using ImageJ software (NIH, Bethesda, MD, USA, http://imagej.nih.gov/ij/, accessed on 22 October 2018)). Amplified PCR products were also electrophoresed and semiquantified using a DNA 1000 LabChip Kit or a DNA 7500 LabChip Kit on an Agilent 2100 Bioanalyzer (Agilent Technologies, Santa Clara, CA, USA). The quantification of each band was performed in triplicate. For quantification of transcripts, each band was corrected for *GAPDH*. Each value was then normalized by the control value.

### 4.4. ASOs

The ASOs were 18-mer-long single-stranded nucleotides consisting of chimeras of ENA and 2’-OMeRNA and were synthesized by KNC Laboratories Co., Ltd. (Kobe, Japan).

### 4.5. Expression of SRSF5

The plasmids to express SRSF5 were constructed by inserting the cDNA of SRSF5 (1529 bp, NM_006925.3) into the mammalian expression vector pcDNA3.1(+) with a CMV promoter (Thermo Fisher Scientific, Inc.). These constructs were synthesized by FASMAC Co., Ltd. (Atsugi, Japan), and their sequences were confirmed by sequencing. Plasmids carrying SRSF5 sequences and ASOs were transfected into CRL-2061 cells. Cells (2 × 10^5^) were cultured on 12-well culture plates for 24 h. The cells were washed with phosphate-buffered saline and cultured in 800 μL of Opti-MEM. A quantity of 2 μL of 50 μM ASO was incubated with 0, 0.01, 0.1, 0.25, 0.5, or 1 μg of SRSF5 expression plasmid, 4 μL of Lipofectamine 3000, and 2 μL of P3000 in 200 μL of Opti-MEM for 15 min. The ASO-lipid and plasmid-lipid complexes were added to each well of cells. Three hours later, the medium was replaced with RPMI 1640 medium supplemented with 10% FBS and 1% antibiotic-antimycotic solution. After incubation for 21 h, the cells were rinsed twice with phosphate-buffered saline and then collected using the Lysis/Binding Buffer of a High Pure RNA Isolation Kit. The isolated RNA was used for RT–PCR, as well as transcript analysis.

### 4.6. Western Blotting

Cells (2 × 10^5^) grown on 12-well culture dishes were transfected with ASO at a final concentration of 100 nM in 4 μL of Lipofectamine 3000 (Thermo Fisher Scientific, Inc.). After incubation for 48 h, the cells were harvested and lysed in Cell lysis buffer (Cell Signaling Technology, Inc., Danvers, MA, USA) containing protease inhibitors. After incubation on ice for 10 min, the lysates were centrifuged at 12,000× *g* for 10 min to remove insoluble material. The protein concentrations of the cell lysates were determined using QUBIT Protein Assay Kits (Thermo Fisher Scientific, Inc.).

Lysates of ASO-transfected cells were analyzed by Western blotting. Briefly, lysates containing 20 μg of protein were mixed 3:1 with Laemmli Sample Buffer (Bio–Rad Laboratories, Inc., Hercules, CA, USA) and boiled for 5 min. These samples and protein size markers (Protein Ladder One Plus, triple-color for SDS-PAGE; Nacalai Tesque, Inc.) were loaded onto MINI-PROTEAN TGX Precast Gels 4–20% (Bio–Rad Laboratories, Inc.). Following electrophoresis, the proteins were transferred to PVDF transfer membranes (Trans-Blot Turbo Mini PVDF Transfer Packs; Bio-Rad Laboratories, Inc.). The membranes were blocked with 2% ECL Prime Blocking Reagent (GE Healthcare, Little Chalfont, UK). The primary antibody reaction was performed overnight using mouse monoclonal antibodies against a synthetic peptide corresponding to amino acids 1–300 of the N-terminal of human myostatin (ab236511, Abcam, Cambridge, UK) at a dilution of 1:5000. An actin antibody (C4, Santa Cruz Biotechnology, Inc., Santa Cruz, CA, USA) was also used at a dilution of 1:5000. The secondary antibody reaction was performed for 1 h using anti-mouse IgG (HRP-Linked Whole Ab Sheep, NA931-1ML, Cytiva, Marlborough, MA, USA) at a dilution of 1:10,000. Immunoreactive bands were detected with ECL Select Western Blotting Detection Reagent (Cytiva) using a ChemiDoc XRS Plus imaging system (Bio–Rad Laboratories, Inc.).

### 4.7. Myostatin-Signaling Assay with SMAD-Responsive Luciferase

Myostatin-signal activity was assayed using the SBE4-Luc luciferase reporter gene containing the SMAD-binding sequence (CAGA) (#16495, Addgene, Watertown, MA, USA). Cells were seeded at a density of 2 × 10^5^ cells/well in a 12-well culture plate for 24 h before transfection with the SBE4-Luc luciferase reporter gene, a pSV-β-galactosidase control vector (#E1081, Promega, Madison, WI, USA), and either ASO or water. The cells were washed with phosphate-buffered saline and cultured in 800 μL of Opti-MEM. The ASO at final concentrations of 0, 1, 5, 10, 25, 50, 100, and 200 nM was incubated with 3 μg of the SBE4-Luc luciferase reporter gene, 3 μg of the pSV-β-galactosidase control vector, 4 μL of Lipofectamine 3000, and 12 μL of P3000 in 200 μL of Opti-MEM for 15 min. The ASO-lipid and plasmid-lipid complexes were added to each well of cells. Three hours later, the medium was replaced with RPMI 1640 medium supplemented with 1% antibiotic-antimycotic solution. After 21 h, the cells were processed, and their luciferase and β-galactosidase activities were measured with a microplate reader (ARVO X3, PerkinElmer, Waltham, MA, USA) using a luciferase reporter assay system and a β-galactosidase enzyme assay system, respectively (Promega). All experiments were performed in triplicate. The values were normalized to β-galactosidase activity and are expressed in relative luminescence units (RLU), which were calculated with the following equation: RLU = luminescence/activity of β-galactosidase.

### 4.8. Cell Proliferation Assays

CCK-8 assay. Cells (5 × 10^3^/well) were plated in triplicate in 96-well culture plates and cultured for 24 h. A quantity of 2 μL of 5 μM ASO was incubated with 0.4 μL of Lipofectamine 3000 in 100 μL of Opti-MEM for 15 min. The cells were washed with phosphate-buffered saline and treated with 100 μL of the ASO-lipid complex. Three hours later, the medium was replaced with DMEM supplemented with 20% FBS, 2% Ultroser G, and 1% antibiotic-antimycotic solution. For the CCK-8 assay (Dojindo, Rockville, MD, USA), 10 μL of CCK-8 solution was added to each well, and the cells were incubated for 1 h at 37 °C in a 5% CO_2_ humidified incubator at 0, 1, 2, 3, and 5 days after transfection. The absorbance of each well at 450 nm was measured with a microplate reader (ARVO X3, PerkinElmer). The results shown are representative of three identical experiments.

Cell number assay. Cell proliferation was analyzed by counting cells under a microscope. Myoblasts (1 × 10^3^/well) were plated in triplicate in 96-well plates. ASO transfection was performed in the same manner as for the CCK8 assay. Cells in a 1 mm^2^ area of each well captured by the 4× plan fluor lens of a BZ-X710 fluorescence microscope (Keyence, Osaka, Japan) were counted after transfection and analyzed using BZ-X analytic software. The same area of each well was analyzed at each time point. The numbers of cells are reported as the averages of three wells containing the same cell populations. The results shown are representative of three identical experiments.

### 4.9. Statistical Analyses

All assays were repeated 3 times to ensure reproducibility. Results reported as the mean ± SE were analyzed by one-way ANOVA and LSD test. Results reported as the means ± SD were analyzed by ANOVA for comparisons of three or more groups and Student’s t-tests for comparisons between two groups. All statistical analyses were performed using SPSS software (version 17.0; SPSS Inc., Chicago, IL, USA), with *p* < 0.05 considered to indicate statistical significance.

## Figures and Tables

**Figure 1 ijms-23-05016-f001:**
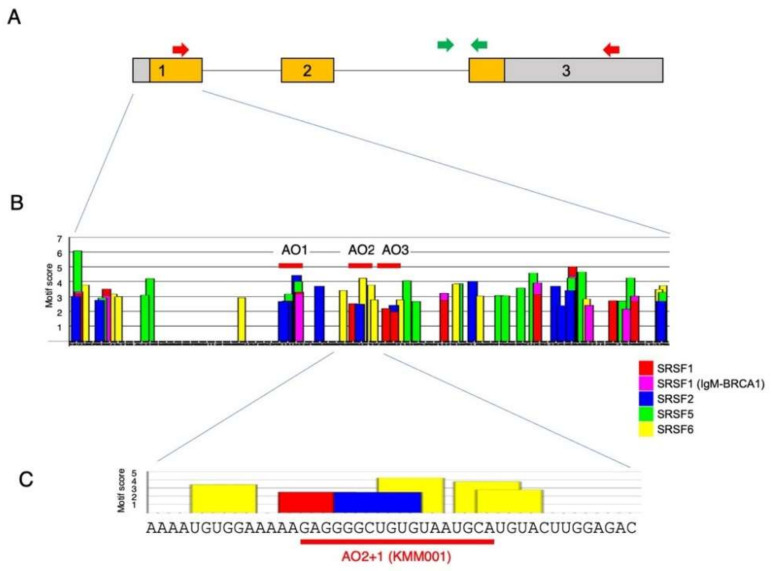
Structure of *MSTN* pre-mRNA and exonic splicing enhancer sequences in exon 1. (**A**) Structure of *MSTN* pre-mRNA. The structure of *MSTN* pre-mRNA is schematically described. The *MSTN* gene comprises three exons and two introns. Exon 1 is 506 nt, and the ATG start codon is present at nucleotide 134. Boxes and lines indicate exons and introns, respectively. The number in the box indicates exon number. The colored area within the box indicates the myostatin-coding sequence. For detection of mature mRNA, a fragment extending from exon 1 to 3 (2328 bp) was PCR amplified using primers on the respective exons (red arrow). To determine the expression of *MSTN* pre-mRNA, a fragment extending from intron 2 to exon 3 (365 bp) was PCR amplified using primers for the respective regions (green arrow). (**B**) Exonic splicing enhancer sequences in exon 1 of the *MSTN* gene. The nucleotide sequence of exon 1 of the *MSTN* gene was analyzed for the presence of splicing enhancer sequences with ESEfinder3.0. A graph of *MSTN* exon 1 produced by ESEfinder3.0 is shown. Only the high-score values (above the selected threshold) are mapped on the output graph. For the color-coded bars, the height of the bar represents the motif score, the width of the bar indicates the length of the motif, and the color of the bar indicates the SR protein (SFRF1, SFRF1 (IgM-BRCA1), SFRF2, SFRF5, and SFRF6 are indicated by red, purple, blue, green, and yellow bars, respectively). Three ASOs (AO1, AO2, and AO3) were synthesized in the first step of screening (red bar). (**C**) Enlarged graph of the areas around sequences complementary to AO2+1 (KMM001). A part of the output graph (**B**) is shown. The bars and their coloring are identical to those in (**B**).

**Figure 2 ijms-23-05016-f002:**
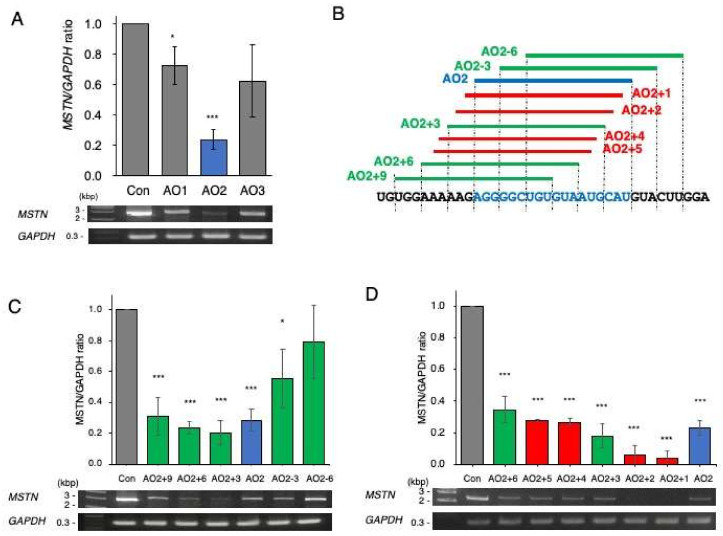
Screening of ASOs that decrease *MSTN* mRNA levels. (**A**) As the first step of screening, three 18-mer ASOs (AO1–3) complementary to the splicing enhancer sequence of exon 1 were synthesized and added to the culture medium of CRL-2061 cells. Mature *MSTN* mRNA was RT–PCR amplified, and the *MSTN*/*GAPDH* ratio was calculated. Electrophoretograms of the RT–PCR-amplified products are shown (**bottom**), and the *MSTN*/*GAPDH* ratios are shown as bars (**top**). All three ASOs decreased the ratio, and AO2 decreased the ratio to approximately 0.2. (**B**). Illustration of the locations of the designed ASOs. To find the ASO that most effectively decreased the *MSTN*/*GAPDH* ratio, another nine ASOs (five (green bar) and four (red bar) ASOs in the second and third screenings, respectively) were synthesized around AO2 (blue bar). The nucleotide sequences are described below, and the sequence complementary to AO2 is colored blue. (**C**). In the second screening, five ASOs (AO2+3, AO2+6, AO2+9, AO2–3, and AO2–6) were synthesized and examined for their ability to decrease mature mRNA levels. Among the five, AO2+3 showed the strongest ability to decrease the ratio. (**D**). In the third screening, another four ASOs (AO2+1, AO2+2, AO2+4, and AO2+5) were synthesized and examined for their suppression ability. Remarkably, AO2+1 decreased the ratio most strongly. Therefore, AO2+1 was selected as the optimal ASO for reducing *MSTN* mRNA levels. (**E**). As the final step, 12 ASOs scattered over exon 1 were synthesized, and their suppression activity was examined. The locations of the ASOs are schematically described (bars) below the exon sequences (the gray and yellow boxes indicate the noncoding and coding regions of exon 1, respectively). The amount of product was calculated and is expressed as the ratio of *MSTN* to *GAPDH*. No ASO showed higher activity than AO2+1. AO2+1 was therefore selected as the best ASO. * = *p* < 0.05, ** = *p* < 0.01, *** = *p* < 0.001 vs. control.

**Figure 3 ijms-23-05016-f003:**
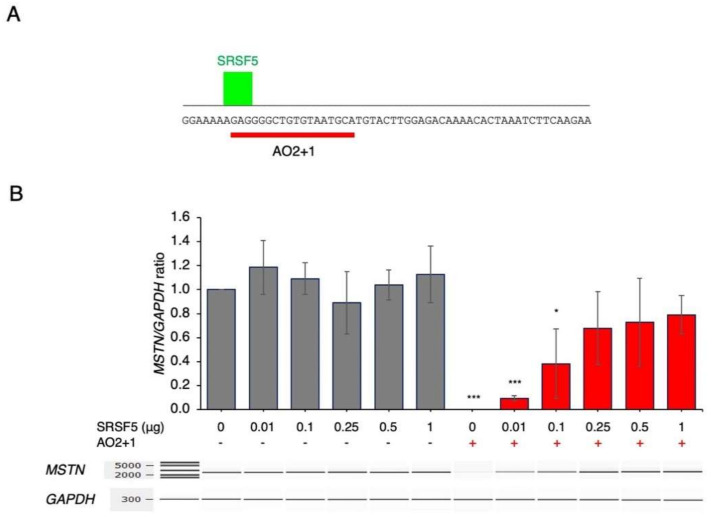
SRSF5 restored the inhibition mediated by AO2+1. (**A**). Characterization of splicing regulatory factors. The nucleotide sequence around the AO2+1 target site was analyzed for splicing enhancer and silencer sequences with SpliceAid2. The output graph is shown. The nucleotide sequences of exon 1 are described in uppercase letters. The binding location of SFRF5 is indicated by a green bar, and the binding strength is represented by the height. A red bar indicates the target site of AO2+1. (**B**). CRL–2061 cells were transfected with different doses of SRSF5 plasmid with or without AO2+1. A fragment extending from exon 1 to exon 3 was RT–PCR amplified. Electrophoretograms of the amplified products are shown (bottom). The amount of product was calculated and is expressed as the ratio of *MSTN* to *GAPDH*. The ratio increased with increasing plasmid dose and reached 0.8 at 1 μg of the plasmid. * = *p* < 0.05, *** = *p* < 0.001 vs. 0 μg SRSF5 and no ASO treatment.

**Figure 4 ijms-23-05016-f004:**
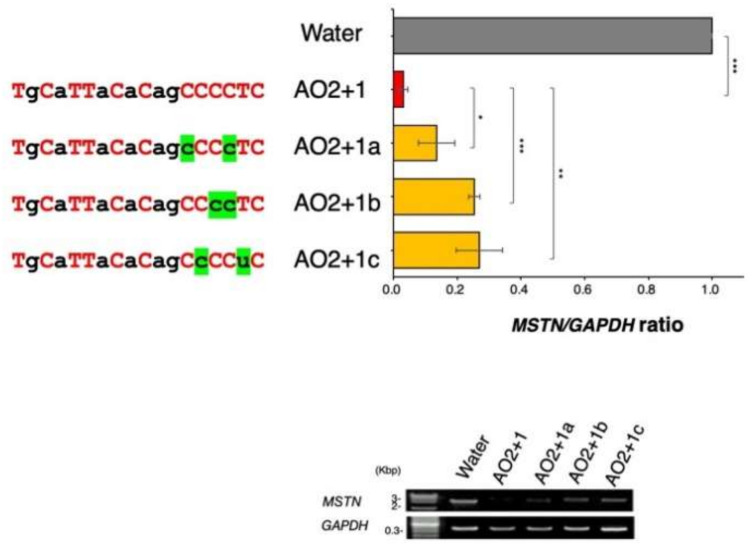
Identification of the best chimeric ASO structure. AO2+1 comprised 12 ENA and 6 2′-OMe RNA components. To find the most effective structure of AO2+1, AO2+1 was modified by replacing ENA with 2′-OMe RNA. Ultimately, three derivatives of AO2+1 were synthesized (AO2+1a, AO2+1b, and AO2+1c). The sequences of these ASOs are described in the top left panel. Red uppercase and black lowercase letters represent ENA and 2′-OMe RNA, respectively. Green marking of 2′-OMe RNA indicates the replaced nucleotide. These ASOs were examined for their ability to inhibit the production of mature *MSTN* mRNA in CRL-2061 cells by RT–PCR amplification (bottom). The amount of product was calculated and is expressed as the ratio of *MSTN* to *GAPDH*. The *MSTN*/*GAPDH* ratio was lowest in AO2+1-treated cells. * = *p* < 0.05, ** = *p* < 0.01, *** = *p* < 0.001 vs. AO2+1 treatment.

**Figure 5 ijms-23-05016-f005:**
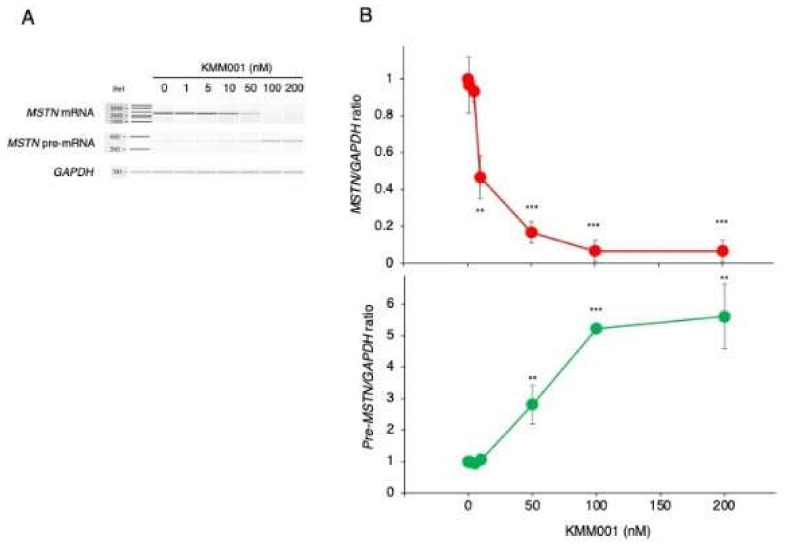
Dose-dependent action of KMM001 on splicing of the *MSTN* gene. (**A**) KMM001 was added to the culture medium of CRL-2061 cells at the indicated concentrations, and the cells were cultured for 1 day. Fragments of exons 1 to 3 and intron 2 to exon 3 of *MSTN* pre-mRNA were RT–PCR amplified. Electrophoretograms of the PCR products are shown. (**B**). The *MSTN/GAPDH* ratio was calculated for each designated concentration after semiquantification of the amplified products using an Agilent Bioanalyzer. The ratio plotted against the concentration of KMM001 is shown (**upper**). The ratio decreased dose dependently from 0 to 100 nM KMM001 and reached its lowest value at 100 nM. Next, the pre-*MSTN/GAPDH* ratio was calculated for each designated concentration. The ratio plotted against the concentration of KMM001 is shown (**lower**). The ratio increased dose dependently from 0 to 100 nM KMM001 and reached a plateau at 100 nM. ** = *p* < 0.01, *** = *p* < 0.001 vs. 0 nM KMM001.

**Figure 6 ijms-23-05016-f006:**
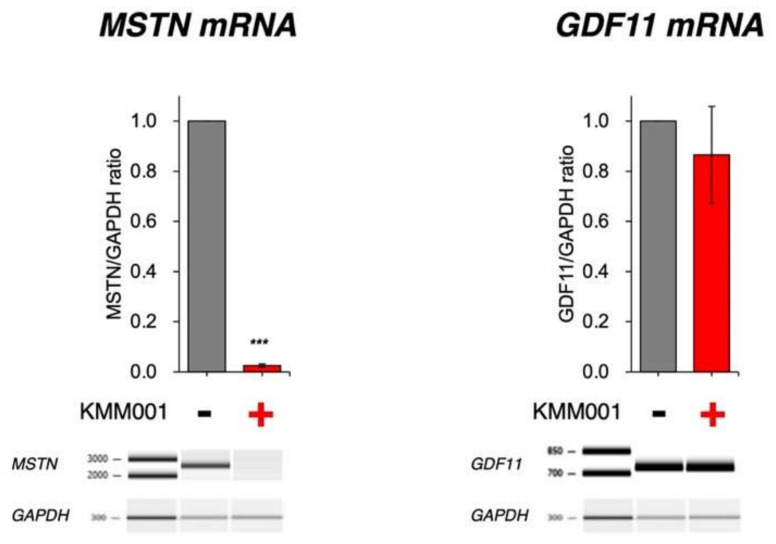
KMM001 did not decrease *GDF11* mRNA levels. To determine the effect of KMM001 on the expression of *GDF11*, CRL-2061 cells were treated with 100 nM KMM001, and the *MSTN* and *GDF11* mRNA sequences were amplified by RT−PCR. Electrophoretograms of the amplified products and their *MSTN/GAPGH* and *GDF11/GAPGH* ratios are shown in the graph. The *MSTN/GAPDH* ratio decreased to 0.03, indicating strong inhibition (*p* < 0.001). On the other hand, the *GDF11/GAPDH* ratio decreased to a level of 0.87 upon addition of KMM001. *** = *p* < 0.001 vs. no treatment.

**Figure 7 ijms-23-05016-f007:**
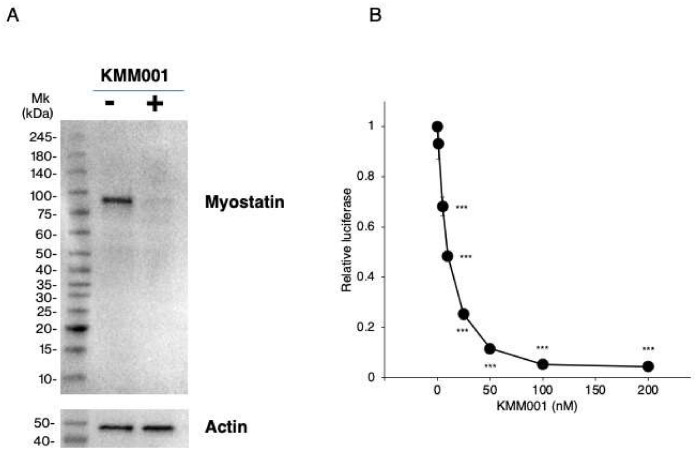
Inhibition of myostatin protein expression and myostatin signaling by KMM001. (**A**). Myostatin protein levels in CRL−2061 cells transfected with AO2+1 were assayed by Western blot analysis using an antibody against the N-terminal domain of human myostatin. SDS–PAGE was carried out under nonreducing conditions. Immunoblotting results are shown. One clear band was identified in untreated CRL−2061 cells (**left lane**). In contrast, in KMM001-treated cells, a band corresponding to myostatin was weakly visualized (**right lane**). Mk refers to the size marker. (**B**). Myostatin signaling was assayed in CRL−2061 cells using the SMAD-dependent luciferase reporter gene. The reporter gene was transfected into cells together with a galactosidase plasmid as a control. The luciferase activity was normalized to the galactosidase activity, and the ratio of luciferase activity to galactosidase activity was calculated. The relative luciferase activity was set to 1 in the nontreated cells. The relative luciferase activity decreased dose dependently from 0 to 100 nM KMM001 and reached a plateau. *** = *p* < 0.001 vs. 0 nM KMM001.

**Figure 8 ijms-23-05016-f008:**
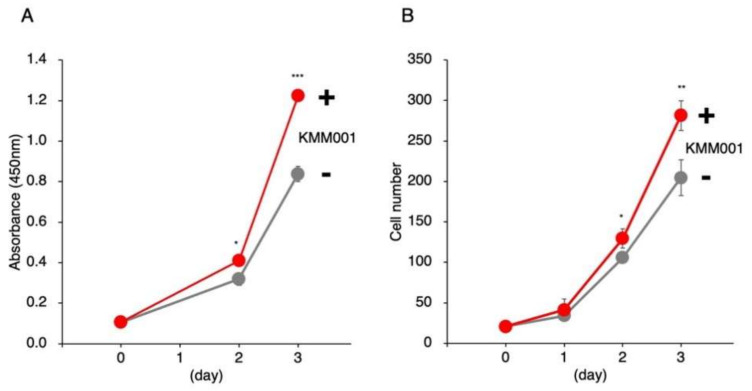
KMM001 enhanced the proliferation of human myoblasts. (**A**). KMM001 was added to the culture medium of immortalized human myoblasts. In a CCK-8 assay, the absorbance increased in treated (red line) and nontreated (gray line) cells. Notably, the absorbance of the cells treated with KMM001 was significantly higher at third day than that of the nontreated cells (146%, *p* < 0.001). (**B**). Direct cell counting using a fluorescence microscope showed that the numbers of cells increased over time in both the treated (red line) and nontreated (gray line) groups. Notably, the number of KMM001-treated cells was significantly higher than that of nontreated cells at third day (138%, *p* < 0.01). * = *p* < 0.05, ** = *p* < 0.01, *** = *p* < 0.001 vs. 0 day.

## Data Availability

All data are available on request.

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
