# Peer review of "An Antisense Oligonucleotide against a Splicing Enhancer Sequence within Exon 1 of the MSTN Gene Inhibits Pre-mRNA Maturation to Act as a Novel Myostatin Inhibitor"

_ijms, 2022, doi:10.3390/ijms23095016_

Round 1

Reviewer 1 Report

Journal: International Journal of Molecular Sciences

Manuscript Number: ijms-1648396

Full Title: An antisense oligonucleotide against a splicing enhancer sequence within exon 1 of the MSTN gene inhibits pre-mRNA maturation to act as a novel myostatin inhibitor.

Comments to the Authors

In the manuscript, Maeta et al. designed and synthesized modified antisense oligonucleotides (ASOs) targeting exon 1 of the MSTN gene and identified one that specifically and efficiently inhibit pre-mRNA splicing of the MSTN gene and confirmed reduction of myostatin protein level in a Rhabdomyosarcoma cell line CRL-2061. They also confirmed that the ASO, termed KMM001, enhanced proliferation of human myoblasts. With these results, the authors claim that KMM001 has great promise for clinical applications and should be examined for its ability to treat various muscle wasting conditions.

The manuscript is clearly written and easy to read. Most of the experiments are straightforward and the results are clear. I have two major concerns. 1) In Figure 8, the authors used human myoblasts that is likely distinct from CRL-2061. If so, reduction of MSTN mRNA and protein levels in the human myoblasts upon treatment with KMM001 should also be confirmed. 2) The authors claim “These results indicate that AO2+1 inhibits splicing by hindering the interaction of SRSF5 with the splicing enhancer sequence.” (lines 182-183). However, restoration of intron 1 excision inhibited by KMM001 upon overexpression of SRSF5 does not necessarily indicates that SRSF5 is involved in the excision in the normal condition. Expression of SRSF5 may have bypassed demand of other factor(s) blocked by KMM001. To claim that SRSF5 is involved in the splicing, the authors should demonstrate that SRSF5 does bind to the MSTN pre-mRNA and KMM001 blocks the binding at least in vitro.

Minor points:

  1. Line 12, 77 and others, “out-of-frame exons”: It seems like the exons are not multiple of three bases in length and skipping of such exons would cause frameshifts. “ASO-mediated out-of-frame exon skipping” would fit the meaning.
  2. Lines 41, 342 and 345, “middle exons”: “internal exons” would fit the context.
  3. Line 78, “to destroy MSTN gene function”: The gene name should be italicised.
  4. Lines 110-128, Legend to Figure 1: Panel titles are missing.
  5. Lines 125, “Three AOs”: “ASOs” would fit the context.
  6. Line 176, “SFSF5”: likely “SRSF5”.
  7. Lines 221, 222 and many others, “linearly”: Not clear how accurately this term is used.
  8. Lines 248-249: “in silico analysis of the GDF11 gene sequence did not reveal any regions that were highly homologous to the complementary sequence of KMM001”: Nucleotide sequence alignment would help readers realize that KMM001 would not bind to GDF11.
  9. Lines 252-253, “KMM001 decreased the MSTN/GAPDH ratio dramatically to 0.03, while the ratio was 1 in nontreated cells”: It seems that this is a normalized ratio and not an absolute ratio. If so, clearly indicate so.
  10. Line 253, “(p<0.001)” and others: Method of the statistical analysis should be clarified.
  11. Lines 253-254, “the addition of KMM001 slightly decreased the GDF11/GAPDH ratio to 0.87 (p=0.29)”: The decrease seems slight and insignificant.
  12. Line 275, “An electrophoretogram is shown.”: It does not seems this is an electrophoretogram.
  13. Line 273, “CRL-2061 cells transfected with AO2+1”: It is described here that KMM001 was transfected into cell, in other context, KMM001 was just added to the culture medium. Please clarify which is the exact method and describe the method in Materials and Methods. “KMM001” should be used instead of “AO2+1” here.
  14. Lines 293-294, “KMM001 was therefore selected as a myostatin signaling inhibitor.”: KMM001 had already been selected before this experiment.
  15. Line 298, “immortalized human myoblasts”: Please clarify the identity of the cells.
  16. Lines 303-304, “the numbers of cells increased over time in both cell lines”: Only one cell line? is used in Figure 8.
  17. Figure 2E, 3B and 6: The y axes likely indicate normalized ratios of MSTN/GAPDH PCR products, but are not explained in such a way in the legends.
  18. Figure 2E, 3B, 4, 5, 6, 7B, 8: Methods for the statistical analyses should be explained in the legend.
  19. Figure 7: It is unclear whether culture medium, cell lysate or both were used in the Western blotting.
  20. Figure 8: Unclear how many replicates were assayed for the figure. The red and gray lines appear the same when printed in black and white. Please use distinct symbols.
  21. Supplementary Figure 1: It should be clarified in the legend whether or not the intron-retention isoform has been identified by direct sequencing or cloning of the RT-PCR products.

Reviewer 2 Report

Maeta et al. tested antisense oligonucleotides (ASOs) against the exon 1 of the myostatin gene in cultured rhabdomyosarcoma cells as well as immortalized human myoblasts. Their study provides a convincing argument that ASOs could be use to suppress myostatin action, which may lead to improved skeletal muscle mass and function.

Comments:

  1. The phrase “to destroy gene function” is used frequently. Perhaps it would be better to say suppress or inhibit? The gene per se is not destroyed.
  2. Abstract: since ASOs are used for different purposes it would perhaps be worthwhile to mention that suppression of gene function is only one of their effects or applications (as nicely mentioned later in the introduction).
  3. Introduction: The authors should check whether previous studies tested whether ASOs induced skipping of a functional exon 2 or an out-of-frame exon 2 of the MSTN. Probably a functional MSTN mRNA was targeted, not an out-of-frame mRNA? (Lines 77-80.) Also, is skipping of out-of-frame exons intended to produce a truncated, but still functional protein (i.e. conversion of an out-of-frame mRNA into an in-frame mRNA), or to suppress it completely (“to destroy gene function”)? Perhaps some sentences should be rephrased to clarify the intended meaning. Indeed, while it is stated in the Discussion that “ASO treatment has been shown to inhibit splicing of exon 2 of the MSTN gene, thereby inducing exon skipping to produce out-of-frame mRNA [22]”, the Introduction refers to: “ASOs that can induce skipping of out-of-frame exon 2 of the MSTN gene in order to destroy MSTN gene function have been studied [22, 78 23].” Since both statements refer to the same study, they probably cannot be true at the same time.
  4. Methods:
    1. What was the reason for using RPMI as well as DMEM for growing CRL-2061 cells?
    2. What kind of DMEM (normal or high glucose?) was used for growing CRL-2061 cells and human myoblasts.
    3. Were antibiotics/antimycotics present in media during experiments (e.g. when assessing proliferation)?
  5. Results:
    1. It would be better if Figure 2 were not split on two pages, while Figure 5 would be easier to read if it were larger.
    2. Detailed figure legends are always useful, but the authors may whish to consider shortening them. Too much text reduces the effectiveness of a figure legends. E.g. legend to Figure 2 has approximately 300 words.
    3. While some information may not be necessary in a figure legend, it would be important to include basic statistical information (the number of measurements, independent experiments, etc.) so that the reader does not need to search for this in the methods.
    4. The authors nicely show that the expression of GDF11 mRNA was not markedly affected by KMM001. However, is it possible that KMM001 would suppress translation, thus leading to reduced GDF11 protein expression? It would be nice if a blot of GDF11 could be shown in addition to the myostatin blot.
    5. The expression of mRNA was assessed by quantifying amplicons on a gel. Is it possible that a more sensitive method, such as real-time PCR, would show a reduction in the GDF11 mRNA levels? It would be nice if a real-time PCR analysis (at least for estimation of GDF11 mRNA) could be included in the manuscript.
    6. Results show that silencing MSTN increases myoblast proliferation, which may be interpreted as an improvement of regeneration capacity. However, on the other hand, uncontrolled proliferation may reduce the capability of myoblasts to form myotubes. Was the myotube formation normal? Also, do authors have data regarding effect of MSTN silencing on the expression of myogenic markers? It would be nice if authors could provide some comments with regard to these questions and, ideally, provide some experimental data.
    7. CCK-8 test showed an increase in the number of myoblasts on the second and third day (Figure 8), indicating an increase in proliferation. However, an increase in cell number could also, at least in part, reflect an increase in viability. Is it possible that the suppression of myostatin increased myoblast viability rather than accelerated their proliferation?
    8. Perhaps it would be better to discuss limitations of the study in the Discussion rather than at the end of the Results.

Round 2

Reviewer 2 Report

Manuscript has been improved by the revision, but there are still some issues that need to be addressed:

  • Since ASOs do not have a single mechanism of action, I still think this fact should be mentioned or taken into the account in the abstract. I think this could be done with only a slight revision.
  • The issue about the skipping of the out-of-frame exon 2 has not been clarified. It is stated in the manuscript that “ASOs that can induce skipping of out-of-frame exon 2 of the MSTN gene in order to suppress MSTN gene function have been studied [22,23].”, but also “ASO treatment has been shown to inhibit splicing of exon 2 of the MSTN gene, thereby inducing exon skipping to produce out-of-frame mRNA [22].” So, was ASO treatment used to skip an out-of-frame exon, which would presumably produce a stable protein, or was it used to produce an out-of-frame exon. The authors should check what they wanted to say. This is important because the mechanism is mentioned also in the abstract: do authors want to say that an out-of-frame mRNA was produced as in the second statement or that an out-of-frame exon was skipped? The authors of ref. 22 actually say: Our results show that the second exon of the MSTN gene can be skipped in multiple myotube cultures, either derived from healthy individuals, DMD patient or mouse. The removal of exon 2 disrupts the open reading frame and introduces premature stop codon.” However, in the abstract of the current manuscript it is stated that “ASOs that induce skipping of out-of-frame exon 2 of the MSTN gene have been studied for their use in increasing muscle mass.”
  • Media may affect the outcome of the experiment. Since two different media were used for CRL-2061 it should be clarified for what purpose or which experiments these tow media were used. Probably both media were used for a specific purpose, which should be mentioned in the methods.
  • Description of the statistics in the Methods is useful; however, it does not tell the reader what is being compared on a specific figure – for instance, are all comparisons done vs. control or are there also other analyses? I think the figure legend should indicate whether differences were compared vs. controls (or vs. another group of measurements) to avoid ambiguity.
  • Since different ASOs produce their effects through different mechanisms, the possibility that GDF11 translation might be suppressed via steric hindrance (of binding to ribosomes) probably cannot be completely excluded. I still think that a GDF11 blot would be valuable, especially since no change in GDF11 protein would prove that the selected ASO specifically targets myostatin.  
  • If the authors do not have data regarding myoblast differentiation into myotubes or expression of myogenic markers, the issue of proliferation vs. differentiation should at least be discussed in the Discussion. Also, lack of these data should be mentioned in the limitations of the study. As regards viability the authors may wish to consult papers such as: PMID: 11162556.

Round 3

Reviewer 2 Report

The manuscript has been improved by the revision; however, there appears to be a misunderstanding of two of my comments:

  1. First, the authors use the phrase “skipping of out-of-frame exon” in the context of silencing/suppressing the function of myostatin. With regard to this point I wanted to say that this statement might be misunderstood for linguistic reasons at least by some readers.

Indeed, the phrase “to skip an out-of-frame exon” is also used to describe how a functional protein is produced with ASOs. Just to give one example from the literature: “Reading frame restoration can enable the expression of nonfunctional genes, often by reframing or skipping of out-of-frame exons, as is common for Duchenne muscular dystrophy (DMD).

However, in the case of myostatin it is the functional (in-frame) exon that is being skipped in order to suppress its expression. The phrase “skipping of out-of-frame exon” would therefore not be understood in the same sense as “out-of-frame skipping” at least by some authors and readers. I therefore recommend to the authors to avoid “skipping of out-of-frame exon” in the context of myostatin and use an expression, which has an unambiguous meaning. This is just a small change in a few parts of the text. My comment was intended only to help to avoid any possible misunderstanding of the intended meaning.

  1. Second, I am aware that GDF11 data refer to mRNA (the authors will probably recall that during the first round of revision I asked why real-time PCR had not been used to estimate the mRNA levels). Since mRNA levels do not necessarily correspond to the protein levels, while ASOs may potentially interfere with the translation, which could, at least theoretically, reduce GDF11 protein levels, I think it would be valuable to show that GDF11 protein was unaltered. However, if this is not possible, the possibility that ASOs might have some unwanted additional effects like interfering with the efficiency of the translation (e.g. due to heteroduplex formation and steric hinderance), which may affect GDF11 protein levels despite having no clear effect on its mRNA should be at least mentioned in the Discussion.

  1. Finally, precisely because myoblast proliferation and differentiation are two different and to some extent opposing processes, I think the manuscript would benefit if a few sentences were added about the potential effect of myostatin silencing on myogenesis.

Round 4

Reviewer 2 Report

/